# Outcome Prediction of Speech Perception in Quiet and in Noise for Cochlear Implant Candidates Based on Pre-Operative Measures

**DOI:** 10.3390/jcm13040994

**Published:** 2024-02-08

**Authors:** Tobias Weissgerber, Marcel Löschner, Timo Stöver, Uwe Baumann

**Affiliations:** 1Audiological Acoustics, ENT Department, University Hospital, Goethe University Frankfurt, 60590 Frankfurt am Main, Germanyu.baumann@med.uni-frankfurt.de (U.B.); 2ENT Department, University Hospital, Goethe University Frankfurt, 60590 Frankfurt am Main, Germany

**Keywords:** word recognition, speech reception threshold, generalized linear model, CI SPiN outcome

## Abstract

(1) Background: The fitting of cochlear implants (CI) is an established treatment, even in cases with considerable residual hearing but insufficient speech perception. The aim of this study was to evaluate a prediction model for speech in quiet and to provide reference data and a predictive model for postoperative speech perception in noise (SPiN) after CI provision. (2) Methods: CI candidates with substantial residual hearing (either in hearing threshold or in word recognition scores) were included in a retrospective analysis (*n* = 87). Speech perception scores in quiet 12 months post-surgery were compared with the predicted scores. A generalized linear model was fitted to speech reception thresholds (SRTs) after CI fitting to identify predictive variables for SPiN. (3) Results: About two-thirds of the recipients achieved the expected outcome in quiet or were better than expected. The mean absolute error of the prediction was 13.5 percentage points. Age at implantation was the only predictive factor for SPiN showing a significant correlation (r = 0.354; *p* = 0.007). (4) Conclusions: Outcome prediction accuracy for speech in quiet was comparable to previous studies. For CI recipients in the included study population, the SPiN outcome could be predicted only based on the factor age.

## 1. Introduction

In the past decades, the provision of cochlear implants (CI) has become an established therapy for patients without sufficient word recognition with hearing aids (HA) or other acoustic hearing solutions [1,2]. The expansion of indication criteria [2,3,4] with respect to preoperative pure-tone audiograms and aided word recognition scores (WRS) at a conversational level of 65 dB SPL, WRS_65_(HA), and postoperative word recognition with CI, WRS_65_(CI), improved in comparison to a patient population with poorer preoperative hearing [5,6]. Especially in subjects within the transition range between hearing aid and cochlear implant indication, e.g., with pure tone thresholds of around 60 to 80 dB and WRS_65_(HA) scores up to 60%, an individual prediction of expected WRS_65_(CI) is of increasing importance. In this subject population, there is a high demand for outcome prediction with CI during the consultation since they oftentimes struggle if they are potentially still “to good” for cochlear implantation and would potentially perform better with their hearing aids than with CIs.

In first attempts, some studies have shown that the preoperative maximum word recognition score, WRS_max_, is one predicting factor for WRS_65_(CI) [6,7,8]. The WRS_max_ should be exceeded by the WRS_65_(CI) in most CI recipients. This finding is of special importance for individual counselling of CI candidates since the information-carrying capacity WRS_max_ [9] is oftentimes not achieved by using hearing aids [10,11,12,13,14,15,16]. Especially in the transition range between HA and CI indication, i.e., for pure-tone thresholds between 60 and 80 dB, the WRS_65_(HA) is on average only half of the WRS_max_ in a patient cohort typically for a maximum care hospital [13]. Additionally to the WRS_max_, recent studies found [17] and confirmed [18,19] that the age at implantation and the WRS_65_(HA) significantly contribute to the prediction of the WRS_65_(CI). This relation can be characterized as
(1)$${WRS}_{65}\left(CI\right)\left[\%\right]=\frac{100}{1+e^{-\left(\beta_{0}+\beta_{1}\cdot {WRS}_{max}+\beta_{2}\cdot Age+\beta_{3}\cdot {WRS}_{65}\left(HA\right)\right)}}$$
with β_0_ = 0.84 ± 0.18, β_1_ = 0.012 ± 0.0015, β_2_ = −0.0094 ± 0.0025 year^−1^, and β_3_ = 0.0059 ± 0.0026; all WRS scores are expressed in %. The application of this generalized linear model (GLM) results in a prediction error (median absolute error, MAE) of 13.5 percentage points in CI recipients with a preoperative WRS_max_ larger than zero. If this measure for outcome prediction is implemented in the postoperative process, a possible mismatch between the reached and predicted WRS_65_(CI) can have an impact on the clinical aftercare process, leading to a further decrease in the MAE down to 11.5 percentage points in this patient group [18]. Reliable prediction allows for the early identification of cases with unexpectedly poor speech perception and the start of early intervention within basic and follow-up therapy. After pathophysiological causes and technical malfunctions were excluded, CI sound processor adjustments [20], intensification of therapies, review of user behavior [21,22], and appropriate counseling [23] must be considered.

To summarize, the outcome prediction for WRS_65_(CI) has reached a level of reliability at which it can be used in both preoperative counseling of CI candidates [17,18] and postoperative process management [18,19]. The restriction to a patient group with preoperative WRS_max_ (which is in Germany around 2/3 of most recent CI provisions [18]) led to considerable progress in outcome prediction for WRS_65_(CI). However, to our knowledge, a similar useful outcome prediction for speech perception in noise (SPiN) based on preoperative routine data has not been established, neither in research nor in clinical routine. Additionally, CI indication criteria for WRS in quiet were formulated in all recent guidelines, e.g., [2,4,24], whereas no reliable data are yet available for speech perception in noise. However, speech perception in noise is oftentimes the main problem in everyday life. This is the main reason to ask about possible options for hearing improvement during patient counseling.

Consequently, there is a need to provide more reference data for SpiN after CI provision. Therefore, the goal of this study is to provide reference data and a predictive model for postoperative SpiN. To do so, the approach from Hoppe et al. [17,18] was used, including only CI candidates with a preoperative WRS_max_ larger than zero or substantial pure-tone residual hearing. The outcome for WRS_65_(CI) will be compared to the predicted scores [17] to further evaluate the GLM according to Equation (1). Particularly, the SpiN, here assessed as speech reception thresholds (SRT) in noise, will be evaluated. A corresponding GLM will be derived from the data of this study. Finally, both measures will be put into relation.

## 2. Materials and Methods

The clinic records from 2005 to 2022 were analyzed for subjects provided with CI systems of type Nucleus Freedom or later (Cochlear, Sydney, Australia) with a preoperative WRS_max_ larger than zero or a four-frequency (test frequencies: 0.5/1/2/4 kHz) pure-tone average (4FPTA) in air conduction of better or equal 80 dB. For higher reliability of the ipsilateral post-operative free-field measures, subjects with contralateral normal hearing (i.e., single-sided deafness, SSD) were excluded. The indication criteria were fulfilled by 87 subjects (mean age: 58.3 ± 16.6 years); 30 of them used contour electrodes (5 CI24RE(CA), 11 CI512, 14 CI612); 27 subjects were implanted with straight electrodes (12 CI422, 4 CI522, 11 CI622); and 30 subjects were implanted with slim modiolar electrodes (11 CI532, 19 CI632). All study participants were provided with a CI within the current CI Guidelines [2] in Germany.

The 4FPTAs ranged between 53.8 and 102.5 dB HL (mean: 75.0 ± 8.7 dB HL), 70 of the 87 subjects had a 4FPTA better or equal to 80 dB. WRS_max_ ranged between 0% and 90% (mean: 34.0 ± 22.8%); 79 of the 87 subjects (91%) had a WRS_max_ larger than zero. The aided monosyllable score WRS_65_(HA) (measured at 65 dB SPL, hearing aid in ipsilateral ear with contralateral masking, if necessary) ranged between 0% and 90% (mean: 21.7 ± 20.9%).

Study measures used for the prediction model pre-surgery included pure-tone audiometry (4FPTA), unaided speech audiometry in quiet (WRS_max_), and aided speech audiometry in quiet (WRS_65_(HA). Study measures post-surgery were speech audiometry in quiet, WRS_65_(CI), and speech reception threshold in noise, SRT(CI).

All speech scores in quiet were assessed with the Freiburg monosyllable test [25]. Speech perception in noise was assessed with the German matrix test (Oldenburg sentence test, OlSa, [26,27,28]). The speech level was adaptively adjusted to measure the SRT for 50% correct word recognition, while the noise level was kept constant at 65 dB SPL. The speech and noise signals were presented from 0° azimuth. One OlSa list (20 sentences each) was used. Prior to testing, one practice list was presented to the subject to familiarize the subject with the test procedure and the speech material. The test was conducted in closed-set mode. The test subjects indicated the words on a touch-screen monitor. The test was conducted only in a unilateral setting with contralateral blocking of the ear canal and additional ear muffs (if necessary).

All included recipients were able to perform the postoperative speech in quiet test. The speech-in-noise test was completed by two-thirds of the recipients (57 of 87).

## 3. Results

### 3.1. Preoperative Pure Tone and Speech Audiometry

Figure 1a,b relate the pure-tone thresholds to WRS_max_ and WRS_65_(HA). Figure 1c relates both word recognition scores in quiet. About 80% of the CI recipients had a preoperative 4FPTA of 80 dB or less (better). About 9% (8 of 87) of the recipients had a WRS_max_ higher than 60%. However, as illustrated in Figure 1c, these patients were not able to fully utilize this potential information-carrying capacity [9] with HA.

### 3.2. Postoperative Speech Audiometry

Figure 2a–c relate the preoperative word recognition scores WRS_max_ and WRS_65_(HA) to the postoperative scores WRS_65_(CI) and SRT(CI). No correlation was found between WRS_65_(HA) and WRS_65_(CI). A weak but significant correlation was found between WRS_max_ and WRS_65_(CI) (r_spearman_ = 0.226; *p* = 0.036, Figure 2b). There was no correlation between the postoperative SRT in noise and any of the preoperative audiometric measures, neither WRS_max_, WRS_65_(HA) nor the 4FPTA.

Within our study group with preoperative 4FPTA better or equal to 80 dB and/or WRS_max_ greater than zero, the mean improvement in word recognition scores in quiet with CI compared to HA prior to surgery was 45.9 percentage points. An improved WRS of at least 20 percentage points in 86% of all cases was observed. There was only one case with a significant decrease in word recognition after the CI provision.

### 3.3. Prediction for Word Recognition in Quiet

Figure 3a,b show the differences between measured and predicted (according to Equation (1)) WRS_65_(CI) after twelve months. About 63% of the recipients achieved the expected outcome or were better than expected. In about 37% of the cases, there was a difference between measured and predicted scores greater than 20 percentage points (i.e., an outcome that was worse than expected). The error in WRS prediction (MAE) was 13.5 percentage points.

### 3.4. Prediction for Speech Recognition in Noise

A GLM was fitted to the data. In analogy to a previous study [17], WRS_max_, WRS_65_(HA), age, and 4FPTA were tested as predictive variables for SRT(CI). The results of the statistics of the GLM for the four tested predictive variables are summarized in Table 1.

The age at implantation was the only significant contributing factor. Three of the four tested input variables do not significantly contribute to the SRT (Table 1). Consequently, the GLM has to be reduced to the one predicting variable age. This results in a GLM according to Equation (2):(2)$$SRT\left[dB\right]=\eta_{0}+\eta_{1}\cdot Age$$
with η_0_ = −2.8774 ± 0.915 and η_1_ = 0.0438 ± 0.0161. There was a significant correlation between age at implantation and SRT (r_spearman_ = 0.354; *p* = 0.007).

Figure 4a,b show the differences between measured and predicted SRTs. The largest difference was 7.6 dB. In a third of the recipients (34%), the test was not performed. The MAE was 1.3 dB. The prediction covers a range of 3.3 dB (−2.4 to 0.9 dB), while the measured SRTs differ between −4.6 and +6.4 dB.

## 4. Discussion

### 4.1. Speech Recognition Scores in Quiet

The results of this retrospective study strongly support the indication criteria for CI provision according to the German guideline [2] in patients with considerable preoperative word recognition scores up to 60%. In our study group of CI recipients, preoperative maximum word recognition scores of up to 90% were observed. In this group, a mean improvement from WRS_65_(HA) to WRS_65_(CI) of 45.9 percentage points was achieved, with only one case showing a significant decrement. In 86% of all cases, speech perception in quiet with CI improved by at least 20 percentage points compared to the pre-surgery performance with a hearing aid, which is in the order of results described in other studies [29]. There was one outlier with a WRS_65_(HA) of 90% and a WRS_65_(CI) of 40%.

With respect to both the preoperative CI candidacy assessment [16] and the achieved postoperative results, our findings are consistent with those of other studies [7,8,17,18,29]. Overall, the measured WRS_65_(CI) correspond to the predicted WRS_65_(CI) with a MAE of 13.5 percentage points. Our retrospective study confirms the results of Hoppe et al. [17], who reported a MAE of 13.5 percentage points as well.

However, there were also some differences in results compared to other studies observed. The WRS_65_(HA) did not correlate with the WRS_65_(CI), whereas according to Hoppe and coworkers, the preoperative WRS_65_(HA) would explain at least around 5 percentage points of the WRS_65_(CI). The lower number of subjects (*n* = 87) in the present study compared to Hoppe et al. (*n* = 128, [17]) potentially contributed to this different finding. Furthermore, both studies slightly differ in the inclusion criteria. Hoppe et al. [17] included all recipients with 4FPTA ≤ 80 dB, while in our study we included all patients with 4FPTA ≤ 80 dB and/or WRS_max_ > 0%. This resulted in around 20% of the study population showing a 4FPTA poorer than 80 dB. The rationale for this different inclusion criterion was that the 4FPTA was found to be no predictor for WRS_65_(CI) [17]. The results of a recent study [18] also suggest that WRS_max_ is a better predictive variable than 4FPTA.

Rieck et al. [8] found no correlation between WRS_65_(HA) and WRS_65_(CI) but between WRS_80_(HA) and WRS_65_(CI). They included all adult CI patients in their analysis. There is no contradiction in the different study results since a clear and restricted definition of the study population with respect to their preoperative characteristics seems to be a key element for better outcome prediction.

### 4.2. Prediction Model for Speech Perception Threshold in Noise

Up to now, guidelines for CI candidacy typically only refer to speech in quiet scores [2,4,24], with some including pure-tone thresholds as a criterion [4,24]. Even though some clinics in CI candidacy evaluation already refer to SpiN performance, e.g., [4], this is not common clinical practice yet. The availability of a predictive model for SpiN could have an effect on both the preoperative candidacy evaluation and patient counseling. Furthermore, a model could impact the post-operative evaluation of hearing performance in such a way that, for example, hearing therapy is continued or intensified if hearing success is too poor compared to the prediction. As speech perception in noise is usually the main problem in everyday life for people with hearing loss, this is often the reason for them to ask about possible options for hearing improvement. Accordingly, they would also like to obtain some information during the consultation about how much SpiN could potentially be improved by the intervention. This is of special importance for subject groups with substantial residual hearing and speech perception in quiet, since they often struggle to decide on a cochlear implant due to the risk of losing their residual hearing.

To our knowledge, no model predicting SpiN (SRTs) after CI provision based on preoperative data has been published. Therefore, the aim of this work was to find a prediction model for SpiN based on pre-operative measures. A particular challenge here is that in this population, speech perception pre-surgery is usually not sufficient, such that SRTs in noise (i.e., 50% speech perception in noise) could be determined, and this potentially valuable input variable for the predictive model is not available.

The recipients in which the SRT was measured (two-thirds of all subjects) showed SRTs within a range of −4.6 to 6.4 dB. This range is comparable to the data published by Kießling et al. [30], where 75% of the CI users who were able to perform the OlSa showed SRTs of 1.3 dB SNR or lower (i.e., better). In the presented study, the SRTs of the tested subjects were better or equal to 1.3 dB SNR in 81% of the cases.

In the investigated cohort of CI recipients, certain SRTs can be expected (see Figure 4). For the SpiN prediction with CI, the regression analysis yielded a GLM with the prediction variable age, which was the only variable contributing to the SpiN prediction out of the four tested input variables: WRS_max_, WRS_65_(HA), 4FPTA, and age. The GLM for SRT prediction according to Equation (2) resulted in a MAE of 1.3 dB. This prediction is just slightly larger than the test-retest reliability of the Oldenburg matrix test [28].

It seems rather disappointing that the factors that already proved valuable for speech in quiet prediction do not contribute to the GLM for SpiN but only age. However, the weak correlation (r = −0.37; *p* = 0.004) between post-operative WRS_65_(CI) and post-operative SRT is in accordance with this finding, showing that even after CI provision, speech performance in quiet (at least assessed with the Freiburg monosyllable test) is no reliable predictor for speech perception in noise. Cognitive deficits potentially associated with age appear to play a greater role for speech perception in noise [31,32] than for speech perception in quiet [10,13,17]. This is in line with results from Füllgrabe et al. [33] and Weissgerber et al. [34], showing a significant impact of age on speech perception even in subjects with normal hearing [33] or subjective normal hearing [34]. Weissgerber et al. reported a significant correlation between age and SRT (r = 0.539, *p* < 0.001), which was still significant after partialing out a potential high-frequency hearing loss (r = 0.44, *p* = 0.03).

### 4.3. Limitations of this Study

It must be noted that Hoppe and coworkers used post-operative data assessed 6 months after CI surgery to fit their prediction model, whereas in the present study, data obtained 12 months after surgery were compared with the prediction. The reason was that an assessment of post-operative data already at six months had potentially resulted in an even higher number of not measurable SRTs in noise. However, the observed difference between the median WRS_65_(CI) six months (70%, [17]) and 12 months (75%, present data) after surgery was only 5 percentage points. This corresponds to a study by Holden et al. [35], which concludes that on average, about 90% of the final performance is reached after 6.3 months.

The implementation of a prediction model for SRTs in noise was restricted to only two-thirds of the patient population. For the remaining one-third of subjects, there was no SRT data after surgery available. The main reason for this is probably that the speech perception in quiet was not good enough (either measured or expected) to allow reliable convergence of the SRT to 50% speech perception. This quite high number of subjects without SRT data could also be due to the fact that the present retrospective study analyzed patient data dating back to 2005. Other studies, including more recent CI candidates with a typically better expected outcome than in the last decades, e.g., due to the shorter duration of deafness and oftentimes better residual hearing, found that the speech-in-noise test could be performed in 78% of the study population [36]. The fact that not all recipients were able to perform adaptive measurements of SRT resulted in the clinical practice of assessing speech-in-noise abilities using tests at fixed signal-to-noise ratios (typically 0 or 10 dB) [37]. This practice was discontinued by many clinics as the results improved and more patients were able to perform adaptive SRT measurements. Another aspect for predicting SpiN performance could be to use a test procedure with simpler test material as a measure, e.g., the digits-in-noise test [38]. Maybe simpler test procedures in noise could also be used as a measure and potential predictor pre-surgery in cases of sufficient residual hearing.

In the present study, the age at the date of surgery was found to be a predictor of SRT performance after CI provision. Since speech perception in noise decreases with age in even normal hearing subjects, the predictive model could be adjusted to include the age at intended testing (e.g., 6, 12, 24 months after CI provision) for performance prediction in future models. The predictive model could also be extended, including the etiology and duration of deafness as potential predictors, as these parameters were not assessed in the present work. However, etiology is oftentimes unknown, and, therefore, only a subpopulation could potentially benefit from etiology as a predictive factor. Czurda and coworkers investigated the impact of the etiology and duration of hearing loss on WRS_65_(CI) [39]. In 60% of the 601 analyzed ears, they reported that the etiology was unknown. For the remaining subjects, they showed that etiology had a significant impact on WRS65(CI). The largest negative deviations between measured and predicted WRS65(CI) were found for the etiologies of perinatal asphyxia, Menière’s disease, and trauma, with perinatal asphyxia showing the highest rate of cases (33%) missing the prognosis by more than 20 percentage points. It could be assumed that subjects with these etiologies could expect lower outcomes in WRS65(CI) than the average CI user. On the other hand, superior outcomes (i.e., better than the outcome prediction) were found in subjects with genetic hearing loss, hearing loss, and otosclerosis. In the same study, subjects were divided into two groups with a duration of hearing loss of more or less than 20 years. No significant difference in WRS_65_(CI) was found between these two subgroups. Hoppe and co-workers included “duration of hearing impairment” and “duration of unaided hearing impairment” as model input variables for the prediction of WRS65(CI) [29]. In the subgroup of subjects with WRS_max_ greater than zero, the inclusion of both variables did not result in a lower prediction error of WRS65(CI).

## 5. Conclusions

The word recognition in quiet outcome in the presented study strongly supports the results found in the previous studies for outcome prediction after cochlear implantation. Different from other studies, the word recognition scores with a hearing aid prior to surgery had no impact on outcome after CI provision. For most of the CI recipients in the included study, speech perception in noise could be predicted only based on the factor age.

## Figures and Tables

**Figure 1 jcm-13-00994-f001:**
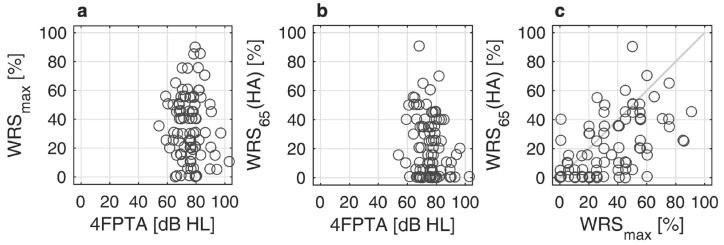
Scatterplots of preoperative pure-tone audiometry and speech audiometry: (**a**) Maximum word recognition score, WRS_max_, as a function of pure tone average, 4FPTA; (**b**) Aided word recognition score, WRS_65_(HA), as a function of 4FPTA; (**c**) Relation between WRS_65_(HA) and WRS_max_.

**Figure 2 jcm-13-00994-f002:**
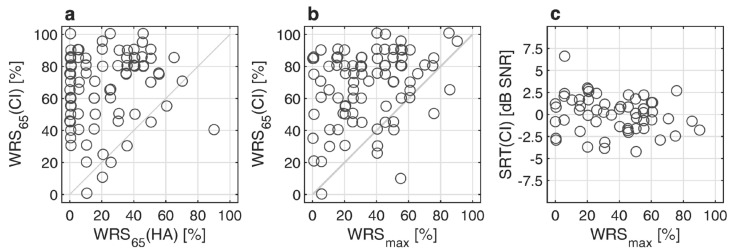
Postoperative pure-tone and speech audiometry after twelve months: (**a**) Word recognition score with CI, WRS_65_(CI), as a function preoperative aided word recognition score WRS_65_(HA); (**b**) WRS_65_(CI) as a function of maximum word recognition score, WRS_max_; (**c**) Relation between speech reception threshold in noise, SRT(CI), and WRS_max_.

**Figure 3 jcm-13-00994-f003:**
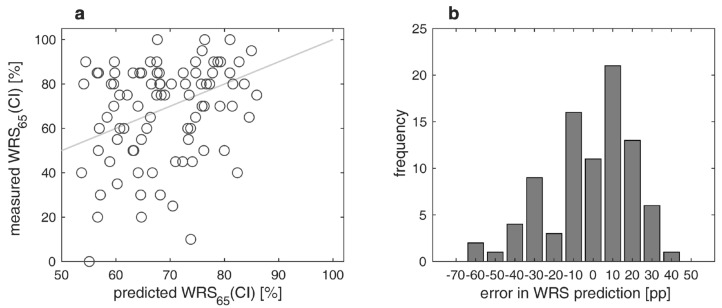
Measured versus predicted word recognition scores in quiet twelve months after CI surgery: (**a**) Relation between measured and predicted scores with CI, WRS_65_(CI); (**b**) differences between measured and predicted WRS_65_(CI). Negative/positive values correspond to poorer/better word recognition than predicted.

**Figure 4 jcm-13-00994-f004:**
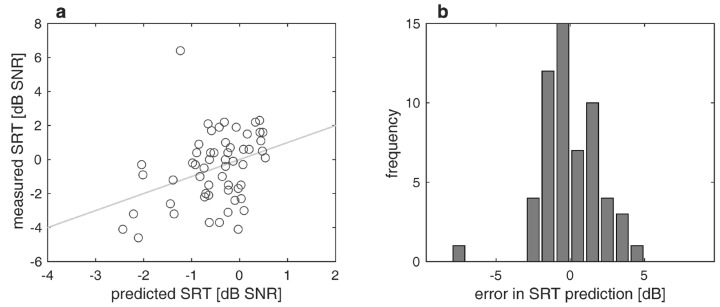
Measured and predicted speech recognition thresholds in noise (SRT) twelve months postoperatively: (**a**) Relation between measured and predicted SRT; (**b**) differences between measured and predicted SRT; negative/positive values correspond to poorer/better word recognition than predicted.

**Table 1 jcm-13-00994-t001:** Results of the regression analysis (generalized linear model, GLM) with the four tested input variables (WRS_max_, WRS_65_(HA), age, and 4FPTA) based on twelve months data for SRT estimation.

	Estimate	Standard Error	T Statistics	*p*
Constant, η_0_	−0.7434	2.6145	−0.2843	0.7772
WRS_max_	−0.0030	0.0141	−0.2180	0.8282
WRS65(HA)	−0.0102	0.0147	−0.6911	0.4925
age, η_1_	0.0408	0.0177	2.2976	0.0256
4FPTA	−0.0213	0.0311	−0.6857	0.4959

Included are 57 observations with 52 degrees of freedom. F-statistics vs. constant model: 3.96, *p* = 0.0949.

## Data Availability

Supporting raw data may be obtained through a special request from the corresponding author.

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
