# Peer review of "Outcome Prediction of Speech Perception in Quiet and in Noise for Cochlear Implant Candidates Based on Pre-Operative Measures"

_jcm, 2024, doi:10.3390/jcm13040994_

Round 1

Reviewer 1 Report

Comments and Suggestions for Authors

Thank you for the opportunity to review this interesting manuscript with Outcome prediction of speech perception in quiet and in noise for cochlear implant candidates based on pre-operative measures.

Abstract: Clear and well written.

In line 14 – please write the word “month” not only “m”.

Introduction: Introduction is a bit difficult to read as it contains complicated calculations, but it provides a basis for what was investigated in the study.

In line 28, 39, 43, 44 – error with the bookmark.

In line 31 – first time use of abbreviation with WRS – explanation is missing. Add this to line 30 to clarify.

In line 33 – is word “borderline” right word?

Material and Methods: Well written and detailed description of the retrospective study in the materials and methods for this study.

Results: Interesting study that shows that the age at implantation was the only significant contributing factor.

Unclear in line 134-135 and in Figure 2: “A significant correlation was found between WRSmax and 134 WRS65(CI) (RSpearman=0.226; p=0.036)” – is it shown in a or b in Figure 2?

In line 170 – what is MAE?

Discussion: The discussion is well discussed and written. The study also discusses well limitations of the study.

Conclusions: Good summary of the study conclusions.

References: Good and relevant references both from earlier and recent publications.

Comments on the Quality of English Language

Good English Language, only minor editing.

Author Response

We would like to thank the reviewer for their positive assessment of our work and their valuable suggestions for improving the quality of the manuscript. Please find attached the point-by-point response to the individual comments

Reviewer 1:

Thank you for the opportunity to review this interesting manuscript with Outcome prediction of speech perception in quiet and in noise for cochlear implant candidates based on pre-operative measures.

Abstract: Clear and well written.
Introduction:
 Introduction is a bit difficult to read as it contains complicated calculations, but it provides a basis for what was investigated in the study.
Material and Methods: Well written and detailed description of the retrospective study in the materials and methods for this study.
Results: Interesting study that shows that the age at implantation was the only significant contributing factor.
Discussion: The discussion is well discussed and written. The study also discusses well limitations of the study.
Conclusions: Good summary of the study conclusions.
References: Good and relevant references both from earlier and recent publications.

In line 14 – please write the word “month” not only “m”.
Changed as suggested.

In line 28, 39, 43, 44 – error with the bookmark.
Bookmarks were corrected

In line 31 – first time use of abbreviation with WRS – explanation is missing. Add this to line 30 to clarify.
The abbreviation was now introduced.

In line 33 – is word “borderline” right word?
Changed into “subjects within the transition range between hearing aid and cochlear implant indication”

Unclear in line 134-135 and in Figure 2: “A significant correlation was found between WRSmax and 134 WRS65(CI) (RSpearman=0.226; p=0.036)” – is it shown in a or b in Figure 2?
This statement is related to Figure b. This was now clarified in the text.

In line 170 – what is MAE?
MAE is the median absolute error. This abbreviation was introduced in line 49. We now added “error in prediction” in line 170 beforehand for better readability and clarity in the manuscript, since this is the first time the abbreviation was used since line 49.

Reviewer 2 Report

Comments and Suggestions for Authors

Very interesting work!

I suggest simplifying the description of the statistical part to improve the reader's comprehension.

Furthermore, why did the authors consider age at surgery as the only predicting factor of WRS? I suppose that all subjects included were post-lingual deaf, but why did not they evaluate also years of deafness, etiology etc?

Author Response

Reviewer 2:

Very interesting work!

We thank the reviewer for the positive assessment of our work and the valuable suggestions for improving the quality of the manuscript. Please find attached the point-by-point response to the individual comments

I suggest simplifying the description of the statistical part to improve the reader's comprehension.
The description of the generalized linear model (GLM) and the output parameters of the model (table 1) in section 3.3 was extended.

 Furthermore, why did the authors consider age at surgery as the only predicting factor of WRS? I suppose that all subjects included were post-lingual deaf, but why did not they evaluate also years of deafness, etiology etc?
This is indeed a valid point. It could be assumed that years of deafness and etiology could be good predictors of outcome after CI surgery. We added this point in the discussion of limitations of the study (4.3):

“The predictive model could also be extended including etiology and duration of deafness as potential predictors as this parameters were not assessed in the present work. However, etiology is oftentimes unknown and, therefore, only a subpopulation could potentially benefit from etiology as predictive factor. Czurda and coworkers investigated the impact of etiology and duration of hearing loss on WRS65(CI) [39]. In 60% of 601 analyzed ears they reported that etiology was unknown. For the remaining subjects they showed that etiology had a significant impact on WRS65(CI). The largest negative deviations between measured and predicted WRS65(CI) was found for the etiologies of perinatal asphyxia, Menière’s disease, and trauma with perinatal asphyxia showing the highest rate of cases (33%) missing the prognosis by more than 20 percentage points. It could be assumed that subjects with these etiologies could expect lower outcomes in WRS65(CI) than the average CI user. On the other hand, superior outcomes (i.e. better than the outcome prediction) were found in subjects with genetic hearing loss, hearing loss, and otosclerosis. In the same study, subjects were divided in two groups with a duration of hearing loss of more ore less than 20 years. No significant difference in WRS65(CI) was found between these two subgroups. Hoppe and co-workers included “duration of hearing impairment” and “duration of unaided hearing impairment” as model input model variables for prediction of WRS65(CI) [29]. In the subgroup of subjects with WRSmax greater than zero, the inclusion of both variables did not result in a lower prediction error of WRS65(CI).”